



# Intercomparison of FY-3 and AIRS Gravity Wave Parameter Extraction Based on Three Methods

Shujie Chang[1, 2, 3], Zheng Sheng[1, 2], Wei Ge[1, 2], Wei Zhang[1, 4], Yang He[1, 2], Zhixian Luo[1]

[1]College of Meteorology and Oceanography, National University of Defense Technology, Nanjing, 210000, China

[2]Collaborative Innovation Center on Forecast and Evaluation of Meteorological Disasters, University of Information Science and Technology, Nanjing, 210044, China

[3]South China Sea Institute for Marine Meteorology, Guangdong Ocean University, Zhanjiang, 524088, 10 China

[4]China Satellite Maritime Tracking and Controlling Department, Jiangyin, 214431, China

[*] Corresponding author.

*Correspondence to*: Zheng Sheng (19994035@sina.com).

**Abstract.** Two types of temperature profile products from the FY-3 (FengYun-3) satellite system, using GNOS and VASS, together with AIRS operational Level 2 data, are used to compare and analyze gravity wave parameters. The advantages and disadvantages of these three types of temperature profile data for gravity wave parameter extraction are determined, based on three extraction methods: vertical sliding average, double-filter and single-filter. By comparing the three methods, the conditions under 20 which each dataset can be applied are obtained. Accurate gravity wave disturbance profiles cannot be obtained using the vertical sliding average method. The double-filter method can extract gravity waves in a vertical wavelength range from 2 to 10 km. The single-filter method can obtain gravity wave disturbances with vertical wavelengths less than 8 km. For all three gravity wave parameter extraction methods, the GNOS temperature profile product performs better in the lower layer of 5–35 km. From 25 35 to 65 km the AIRS temperature profile product is better than GNOS. Using the single-filter method, GNOS and AIRS filter out the vertical linear trend in the disturbance profile well, reflecting the advantages of a single filter. The vertical resolution of VASS is lower, but larger vertical scale components are retained.

Keywords: Gravity Wave; FY-3; AIRS; Extraction Methods.

## 1 Introduction


Atmospheric gravity waves are small-scale or meso-scale disturbances that can propagate vertically (Holton, 1983). Atmospheric gravity waves play an important role in the meteorology, climatology,





chemistry, and dynamics of the stratosphere and mesosphere (Fritts and Alexander, 2003). The

generation of gravity waves depends on topographic and atmospheric conditions: they are primarily

generated due to orography (Smith, 1985; Durran and Klemp, 1987; Nastrom and Fritts, 1992), deep

convection (Pfister et al., 1986; Tsuda et al., 1994; Alexander and Pfister, 1995; Alexander and Vincent,

2000), wind shear (Fritts and Nastrom, 1992; Plougonven et al., 2003; Wu and Zhang, 2004) and

wave-flow interactions (Fritts and Alexander, 2003; Vadas et al., 2003). During their upward

propagation, since the kinetic energy is inversely proportional to the square root of the atmospheric

density, the amplitude of the disturbance increases exponentially. When a critical layer is reached, this

leads to fragmentation, so that the momentum and energy are released into the background atmosphere,

resulting in a change to the background circulation that affects the thermodynamics and dynamics of

the atmosphere both locally and globally (Li and Yi, 2007; Zhang et al., 2010; Tang et al., 2014).

During the whole lifecycle of gravity waves, the occurrence and development of various mesoscale

weather systems are profoundly affected by their generation, development and fragmentation processes,

which are often triggers for various meso-scale convective systems. Gravity waves also play a

significant role in the adjustment and evolution of atmospheric circulation. At the same time, gravity

waves are the link between the lower atmosphere and the middle and upper atmosphere (Holton, 1982;

Lamarque et al., 1996; Sprenger et al., 2007; Pan et al., 2010). At present, most global atmospheric

models use gravity wave parameterizations. The gravity wave parameterization scheme is an essential

module in almost all atmospheric general circulation models (GCMs), including for middle atmosphere

processes (Fritts and Alexander, 2003). Generally speaking, the importance of stratospheric gravity

waves in atmospheric circulation modeling and numerical weather prediction mode has two main

aspects. First, by using data obtained from various observation methods, information about

stratospheric gravity waves can be extracted, and their distribution characteristics analyzed, and this is

necessary for accurately constructing and executing the atmospheric model. Second, considering the

subgrid effects of stratospheric gravity waves is important for constructing the parameterization scheme

itself (Fritts and Alexander, 2003; Kim et al., 2010).

During aircraft flight, since the scale of the gravity waves is similar to the typical aircraft size,

stratospheric gravity waves have a strong influence on the aircraft, and can periodically cause it to

vibrate. After the gravity wave breaks, the turbulent motion generated can also impact the aircraft

motion irregularly but frequently, affecting the flight path and causing safety issues. Because of the

unique environmental characteristics of the stratosphere, any strong disturbances will also bring flight

safety problems to Low Earth Orbit (LEO) satellites (Lane et al., 2003; Williams and Joshi, 2013). In

general, improving the understanding of gravity waves in the stratosphere cannot be neglected. On the

one hand, it is essential for improving the accuracy of atmospheric circulation models and the

numerical weather prediction; on the other hand, it is an urgent requirement for flight safety.



Satellite observation has been widely used in the study of gravity waves. Fetzer and Gille (1994) demonstrated for the first time that satellite remote sensors can observe gravity wave systems.

Subsequently, the global distribution characteristics of stratospheric gravity waves in a given year have been studied by using GPS/MET occultation data (Tsuda et al., 2000), CHAMP GPS occultation data (Ratnam et al., 2004; Torre et al., 2006), Aura satellite MLS (Microwave Limb Sounder) observations (Wu and Eckermann, 2008) and COSMIC GPS occultation data (Xiao and Hu, 2010). Those studies have shown that the distribution of gravity waves changes significantly not only with altitude, but also

with latitude, longitude and topography. In order to construct a more systematic and reliable gravity wave model, Ern et al. (2014) used SABER's 11-year observation data and HIRDLS (High Resolution Dynamics Limb Sounder) two-year observation data to study the contribution of gravity waves to the equatorial quasi-biennial oscillation (QBO). Because of the increasing accuracy, vertical resolution and data density of COSMIC satellite data, current observations of COSMIC satellites are widely used in

the study of short-term atmospheric gravity waves in the global stratosphere (Alexander et al., 2008; Wang and Alexander, 2009; McDonald, 2012). However, there is still a lack of research on stratospheric atmospheric gravity wave activity based on long-term observations of COSMIC satellites. Liang et al. (2014) determined some characteristics of gravity waves in the stratosphere by using the temperature profile data from January 2007 to December 2012.

In 2002, NASA launched the Aqua satellite to combine AMSU (The Advanced Microwave Sounding Unit) with AIRS (The Atmospheric Infrared Sounder), forming a high-resolution coupled temperature detection system. The horizontal resolution is three times higher than that of AMSU-A, enabling AIRS to measure the two-dimensional horizontal structure of gravity waves (Alexander, et al., 2010). Meanwhile, the long-term time series of AIRS radiation data provides further opportunities for studying

the frequency of gravity waves and other climatic features on a global scale (Gong et al., 2012; Hoffmann et al., 2013, 2014). Previously, the temperatures retrieved by AIRS were not usually used for gravity wave studies, mainly because of their limited horizontal resolution. However, a high spatial resolution stratospheric three-dimensional temperature field can be obtained from AIRS radiation Level 1 data. The high-resolution AIRS temperature dataset obtained by Hoffmann and Alexander (2009) is

considered to be the best choice for stratospheric gravity wave research. In the meanwhile, a comparison between the AIRS high-resolution stratospheric temperature retrieval, the AIRS operational Level 2 data, and ERA-Interim reanalyses is performed (Meyer and Hoffmann, 2014; Dee et al., 2011), which    showed that the AIRS high-resolution retrievals reproduce means with good accuracy. Yao et al. (2015) used the AIRS observation data to obtain the characteristics of stratospheric gravity waves in

East Asia in summer. Some study show the estimating directional gravity wave momentum flux which is applied 3-D spectral analysis techniques to the AIRS high-resolution retrievals (Ern et al., 2017; Wright et al., 2017)Gravity wave propagation processes have also been analyzed using AIRS (e.g. Sun





et al., 2018; Wang et al., 2018). Meyer et al. (2018) demonstrated the effectiveness of AIRS for gravity
wave observation by comparing AIRS and HIRDLS temperature profiles. In summary, it is possible to
extract good gravity wave signals from temperature profiles retrieved by AIRS.

It is worth mentioning that Hoffmann and Alexander (2009) found that the minimum vertical
wavelength of their high-resolution AIRS retrieval is about 10-15 km in the stratosphere. For the
temperature retrievals presented, they adapted the Juelich Rapid Spectral Simulation Code
(JURASSIC). The simulations include only AIRS channels where radiance emissions of carbon dioxide
dominate, and contributions of interfering species or aerosols can be neglected in comparison with
noise. Hindley et al.(2019) chosen vertical regular distance grid is over the range $z=10-70$ km in steps
of 3 km, close to the original grid of Hoffmann and Alexander (2009), which comes to the same results.
In order to extract gravity wave signals with a vertical wavelength of less than 10 km, the AIRS
operational Level 2 data is used in this paper. In combination with the Advanced Microwave Sounding
Unit (AMSU) and the Humidity Sounder for Brazil (HSB), AIRS Level 2 data constitutes an innovative
atmospheric sounding group of visible, infrared, and microwave sensors. It is a new product produced
using AIRS IR only because the radiometric noise in AMSU channel 4 started to increase significantly
(since June 2007). The Support Product includes higher vertical resolution profiles of the quantities
found in the Standard Product, plus intermediate outputs (e.g., microwave-only retrieval), research
products such as the abundance of trace gases, and detailed quality assessment information (More
information can be found online at https://disc.gsfc.nasa.gov/datasets/AIRS2SUP_V006). The Support
Product profiles contain 100 levels between 1100 and .016 mb, which will be further introduced in
section 2.

In China, the FY-3C satellite was launched at the Taiyuan Satellite Launch Center on September 23,
2013, at 11:07 am. The FY-3 series is the second generation of polar orbiting meteorological satellites
in China. The goal is to observe the global atmosphere and its geophysical properties on a
whole-weather, multi-spectral and three-dimensional scale. Recently, FY-3 has been used for a
preliminary study on gravity waves (Yao et al., 2019), but its products have not been systematically
applied to gravity wave research. In particular, there has been no research published on how to
optimally analyze the gravity wave parameters using FY-3 products. Therefore, this paper aims to study
the advantages of the FY-3 satellite in obtaining stratospheric gravity wave parameters, in order to
enable FY-3 products to be effectively applied to the study of gravity waves in the future. The two
types of FY-3 temperature profile products, with GNOS and VASS, together with AIRS operational
Level 2 data, are used to compare and analyze gravity wave parameters. The advantages and
disadvantages of these three types of temperature profile data in gravity wave parameter extraction are
determined.



## 2 Data and method

### 2.1 AIRS Level 2 data

The Atmospheric Infrared Sounder (AIRS) on the Aqua in the Earth observation system is designed to measure the Earth's atmospheric temperature profiles on a global scale, which covers a wide band of observation the brightness temperature: 3.74 μm to 4.61 μm, 6.20 μm to 8.22 μm, and 8.8 μm to 15.4 μm, totally in 2378 channel. For studies of atmospheric gravity waves, AIRS radiance measurements are suitable, because it can provide nearly continuous measurement coverage since September 2002. The Level 2 data start from August 31 2002 to present, the height ranging from 0 to 77km. The vertical

profiles of temperature for 100 levels, in km, are shown in Table 1. From 0 to 35km, the vertical resolution can be as low as 1km. From 35 to 50km, the vertical resolution decreases to 2km. Above 55km, the vertical resolution decreases rapidly. It shows that operational Level 2 data can extract gravity wave signals with a vertical wavelength of less than 5 km from 0 to 35km which contains small-scale information.


**Table 1.** Heights adopted for AIRS 100 levels (in km).

| Level number | height km | Level number | height km | Level number | height km | Level number | height km |
|---|---|---|---|---|---|---|---|
| 1 | 77.256844 | 31 | 23.337944 | 61 | 9.0900583 | 91 | 1.1147099 |
| 2 | 71.172173 | 32 | 22.663448 | 62 | 8.7559423 | 92 | 0.90581858 |
| 3 | 66.311028 | 33 | 22.009609 | 63 | 8.4278097 | 93 | 0.69977981 |
| 4 | 62.268707 | 34 | 21.375278 | 64 | 8.105485 | 94 | 0.4965359 |
| 5 | 58.814564 | 35 | 20.759438 | 65 | 7.7887945 | 95 | 0.29603451 |
| 6 | 55.795654 | 36 | 20.161093 | 66 | 7.477581 | 96 | 0.098219335 |
| 7 | 53.117287 | 37 | 19.579374 | 67 | 7.1716838 | 97 | -0.096958578 |
| 8 | 50.712391 | 38 | 19.013428 | 68 | 6.8709564 | 98 | -0.28955156 |
| 9 | 48.529358 | 39 | 18.462509 | 69 | 6.5752563 | 99 | -0.47960705 |
| 10 | 46.532829 | 40 | 17.925901 | 70 | 6.284451 | 100 | -0.66717106 |
| 11 | 44.692795 | 41 | 17.402946 | 71 | 5.9984012 | | |
| 12 | 42.987553 | 42 | 16.893023 | 72 | 5.7169886 | | |
| 13 | 41.399193 | 43 | 16.395557 | 73 | 5.4400892 | | |
| 14 | 39.91291 | 44 | 15.910015 | 74 | 5.1675873 | | |
| 15 | 38.516754 | 45 | 15.435884 | 75 | 4.8993726 | | |
| 16 | 37.200558 | 46 | 14.972706 | 76 | 4.6353397 | | |
| 17 | 35.956245 | 47 | 14.52003 | 77 | 4.3753824 | | |
| 18 | 34.776352 | 48 | 14.077447 | 78 | 4.1194057 | | |
| 19 | 33.654861 | 49 | 13.644554 | 79 | 3.867311 | | |
| 20 | 32.586479 | 50 | 13.220989 | 80 | 3.6190069 | | |



| | | | | | |
|---:|---|---:|---|---:|---|
| 21 | 31.566605 | 51 | 12.806409 | 81 | 3.3744059 |
| 22 | 30.591127 | 52 | 12.400471 | 82 | 3.1334248 |
| 23 | 29.656567 | 53 | 12.002877 | 83 | 2.8959768 |
| 24 | 28.759758 | 54 | 11.613328 | 84 | 2.6619873 |
| 25 | 27.897917 | 55 | 11.231546 | 85 | 2.4313765 |
| 26 | 27.068544 | 56 | 10.857268 | 86 | 2.2040715 |
| 27 | 26.269417 | 57 | 10.490239 | 87 | 1.9799997 |
| 28 | 25.498541 | 58 | 10.130228 | 88 | 1.7590938 |
| 29 | 24.75407 | 59 | 9.7770014 | 89 | 1.5412855 |
| 30 | 24.034367 | 60 | 9.4303474 | 90 | 1.3265123 |

## 2.2 FY-3 temperature profile

The Level 2 data of FY-3 include atmospheric temperature profiles from the Global Navigation

Occultation Sounder (GNOS) and the Vertical Atmospheric Sounder System (VASS) (Liao et al., 2016;

Yao et al., 2019).

GNOS is one of the remote sensing instruments on the FY-3 satellite. The GNOS Atmospheric

Temperature Profile (ATP) products provide the atmospheric moisture profile and auxiliary data for a

single occultation. The products include a record of time, position of perigee point, temperature and

pressure. The data range from June 1, 2014, to the present, and the height ranges from 0 to 65km; this

is generally used as the final atmospheric occultation product (Liao et al., 2016). The vertical profiles

of temperature for 60 levels, in km, are shown in Table 2. From 0 to 35km, the vertical resolution can

be as low as 1km. From 35 to 50km, the vertical resolution decreases to 2km.Above 50km, the vertical

resolution decreases rapidly. It shows that GNOS can extract gravity wave signals with a vertical

wavelength of less than 5 km from 0 to 35km which contains small-scale information. More

instruments information can be found online at http://www.nsmc.org.cn/NSMC/Home/Index.html.

**Table 2.** Heights adopted for GNOS 60 levels (in km).

| Level number | height km | Level number | height km | Level number | height km |
|---:|---|---:|---|---:|---|
| 1 | 63.45001732 | 21 | 21.38025447 | 41 | 4.685979736 |
| 2 | 56.03636747 | 22 | 20.12922932 | 42 | 4.165835684 |
| 3 | 51.94624156 | 23 | 18.94540984 | 43 | 3.67720236 |
| 4 | 48.66232094 | 24 | 17.82621887 | 44 | 3.220883386 |
| 5 | 45.96642879 | 25 | 16.77031706 | 45 | 2.79777204 |
| 6 | 43.6833421 | 26 | 15.76907568 | 46 | 2.40774986 |
| 7 | 41.68826032 | 27 | 14.81382696 | 47 | 2.049966708 |
| 8 | 39.89239239 | 28 | 13.90102507 | 48 | 1.723919046 |
| 9 | 38.23446611 | 29 | 13.0306026 | 49 | 1.429523995 |





| 10 | 36.66612868 | 30 | 12.20142995 | 50 | 1.166710912 |
|----|-------------|----|-------------|----|-------------|
| 11 | 35.14575249 | 31 | 11.40375472 | 51 | 0.934810441 |
| 12 | 33.64531691 | 32 | 10.62818736 | 52 | 0.732656862 |
| 13 | 32.17294806 | 33 | 9.873070715 | 53 | 0.559253834 |
| 14 | 30.75312805 | 34 | 9.138710471 | 54 | 0.413602711 |
| 15 | 29.37799749 | 35 | 8.425806957 | 55 | 0.294295707 |
| 16 | 28.01655026 | 36 | 7.73576406  | 56 | 0.199404957 |
| 17 | 26.6607753  | 37 | 7.070353414 | 57 | 0.126575756 |
| 18 | 25.31667268 | 38 | 6.431115179 | 58 | 0.073026989 |
| 19 | 23.98785926 | 39 | 5.819472775 | 59 | 0.035497364 |
| 20 | 22.67640744 | 40 | 5.237303433 | 60 | 0.010146927 |

The VASS Atmospheric Vertical Profile (AVP) product includes global atmospheric temperature and
humidity profiles retrieved from 4 MWTS (Micro Wave Temperature Sounder) microwave channels, 5
MWHS (Micro Wave Humidity Sounder) microwave channels and a VIRR cloud mask, which has
been matched onto pixels of MWHS. It contains latitude, longitude, land-sea mask, land cover and
surface height, solar zenith angle, solar azimuth angle, satellite zenith angle and satellite azimuth angle
with 98 MWHS pixels per scan line. Meanwhile, brightness temperature of MWTS, brightness
temperature of MWHS, cloud percentage atmospheric temperature and humidity profile on each pixel
are also included in the database. The data range from June 1, 2014, to the present, and the height
ranges from 0 to 65 km; the uses of this product include weather analysis, data assimilation in
numerical weather and climate prediction and research on climate change.

The vertical profiles of temperature for 43 levels, in km, are shown in Table 3. From 0 to 20km, the
vertical resolution can be as low as 1km. From 20 to 30km, the vertical resolution decreases to
2km.Above 30km, the vertical resolution decreases rapidly. It shows that VASS can extract gravity
wave signals with a vertical wavelength of less than 5 km from 0 to 20km.

**Table 3.** Heights adopted for VASS 43 levels (in km).

| Level number | height km | Level number | height km | Level number | height km |
|----|-------------|----|-------------|----|-------------|
| 1  | 64.4723826  | 21 | 11.46486171 | 41 | 0.099402446 |
| 2  | 56.78209658 | 22 | 10.50722427 | 42 | -0.03769831 |
| 3  | 50.85101156 | 23 | 9.601219687 | 43 | -0.09248632 |
| 4  | 45.9989813  | 24 | 8.747674345 |    |             |
| 5  | 41.66570684 | 25 | 7.943304147 |    |             |
| 6  | 37.98305517 | 26 | 7.184692583 |    |             |
| 7  | 34.73291591 | 27 | 6.470260325 |    |             |
| 8  | 31.96164631 | 28 | 5.794754587 |    |             |
| 9  | 29.49189669 | 29 | 5.159693477 |    |             |
| 10 | 27.24554265 | 30 | 4.557322032 |    |             |



| 11 | 25.20608004 | 31 | 3.990314884 |
|----|-------------|----|-------------|
| 12 | 23.36755808 | 32 | 3.453192391 |
| 13 | 21.66113773 | 33 | 2.946894438 |
| 14 | 20.08986748 | 34 | 2.469776545 |
| 15 | 18.61482026 | 35 | 2.022179551 |
| 16 | 17.23927692 | 36 | 1.605011703 |
| 17 | 15.97261788 | 37 | 1.22047371 |
| 18 | 14.72613964 | 38 | 0.872596232 |
| 19 | 13.57532284 | 39 | 0.564675321 |
| 20 | 12.4865391 | 40 | 0.304738014 |


## 2.3 Gravity wave extraction methods

According to the linear theory of gravity waves, the atmospheric temperature profile $T(z)$ typically consists of two components: the background temperature profile $\bar{T}(z)$ and the disturbance part $T'(z)$, which represents the gravity wave: $T(z) = \bar{T}(z) + T'(z)$. Based on vertical filtering, previous research has

adopted the following different processing methods in specific operations. 1: Vertical sliding average method. Using a sliding window with a height of 8 km, the observed temperature profile is averaged to estimate the background temperature profile $\bar{T}$. Then the background temperature profile is subtracted from the original temperature profile to obtain the gravity wave disturbance profile (Hocke and Tsuda, 2001). 2: Double-filter method. On a specific latitude and longitude grid, by taking all the valid

observation data at a given height within 7 days, a weekly average profile of $\bar{T}(z)$ is obtained as the background temperature profile. The disturbance profile can be obtained by subtracting $\bar{T}(z)$ from the observed data. However, deviations occur when subtracting the average profile from a single profile. Therefore, the vertical linear trend is removed. A high-pass filter and a low-pass filter are then used separately to vertically filter out the large-scale and small-scale fluctuations, respectively, thereby

obtaining the gravity wave disturbance $T'(z)$ (Tsuda et al., 2000). 3: Single-filter method. First, the temperature is interpolated in height so that the vertical resolution is 1 km, filtering out small-vertical-scale disturbances and noise. As a result, the vertical wavelength of the extracted gravity wave is less than 2 km. Second, on a specific latitude and longitude grid, the average value of the temperature at each height within a 7-day period is calculated to give the background temperature $\bar{T}(z)$.

Subtracting the background temperature from the original temperature, and removing the vertical linear trend, gives a temperature disturbance profile. Finally, a high-pass filter is used to filter this, to obtain the gravity wave temperature disturbance profile (Alexander et al., 2008).

Theoretically, the first method can extract gravity wave disturbances with a vertical wavelength of less than 8 km, the second method can extract gravity waves in a vertical wavelength range from 2 to 10 km,

and the third method can obtain gravity wave disturbances in the vertical wavelength range from 2 to 8





km. In fact, some planetary scale disturbances, such as Kelvin waves, have vertical wavelengths of the same scale as gravity waves, so using these filtering methods means that the gravity wave disturbances obtained actually include a contribution from these other waves.

Based on the calculated gravity wave disturbances and background temperature, we can further

calculate the square of the buoyancy frequency $N^2$ and the potential energy Ep of the gravity wave:

$$N^2(z) = \frac{g}{\bar{T}}\left(\frac{\partial \bar{T}}{\partial z} + \frac{g}{c_p}\right) \quad (1)$$

$$Ep = \frac{1}{2}\left(\frac{g}{N}\right)^2 \left(\frac{T'}{\bar{T}}\right)^2 \quad (2)$$

where $g = 9.8 \ \mathrm{m \cdot s^{-2}}$, $c_p = 1005 \ \mathrm{J \cdot kg^{-1} \cdot K^{-1}}$, and $\bar{T}$ is the background temperature.

**3 Comparison of FY-3 and AIRS based on three extraction methods**

**3.1 Vertical sliding average method**

Taking the observed temperature profile at (74.65°W, 35.19°N) on January 1, 2019, as an example, the vertical sliding average method is first used to extract gravity wave disturbances from a single temperature profile. The steps are as follows:

(1) Using a sliding window with a height of 8 km and a sliding step length of 500 m, the background

temperature profile $\bar{T}$ is calculated.

(2) The background temperature profile $\bar{T}$ is subtracted from the original temperature profile $T$, to obtain the gravity wave disturbance profile.

Applying the sliding average with a window length of 8 km gives a background containing vertical scales greater than 8 km. Therefore, the gravity wave disturbance obtained by subtracting the

background profile retains those wave components with a vertical wavelength of less than 8 km. For reference, first the calculation results of AIRS only are shown in Fig. 1. Then the calculation results from the FY-3 satellite and AIRS are compared in Fig. 2.



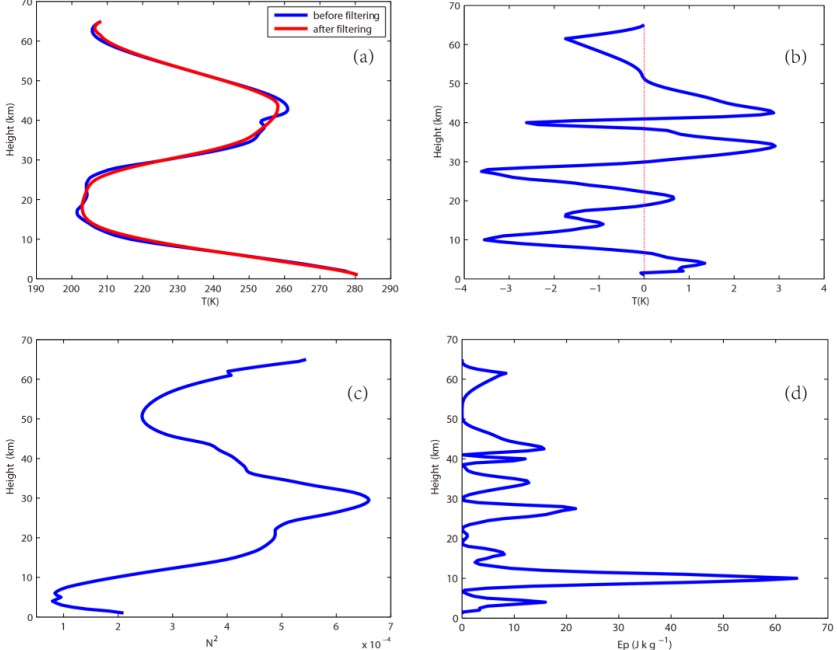

**Figure 1.** Temperature profile observed by AIRS at 74.65ºW, 35.19ºN on January 1, 2019. (a) The original atmospheric temperature profile (blue line) and the average temperature profile (red line). (b) The gravity wave temperature disturbance profile. (c) The gravity wave buoyancy frequency. (d) The gravity wave potential energy. Quantities are calculated using the vertical sliding average method.

The temperature profile observed by AIRS at 74.65ºW, 35.19ºN on January 1, 2019, is given in Fig. 1. Figure 1a shows the original atmospheric temperature profile (blue line) and the average temperature profile (red line), which is calculated using a sliding average with a window length of 8 km. The minimum in the original temperature profile indicates that the height of the tropopause is approximately 18 km, with a temperature of approximately -70 °C. The background temperature shows the variation of temperature with height and gives the height range of the tropopause. Figure 1b shows the gravity wave disturbance profile, which is obtained by subtracting the background temperature profile $\overline{T}$ from the original temperature profile $T$. Due to the complex terrain and inhomogeneous surface, the temperature profile standard deviation of the data sets for heights ranging from 0 to 5 km is large: the temperature profiles here are poor (Liao et al., 2016). Therefore, we only consider the temperature profiles above 5 km in this study.

There are two maxima in the magnitude of the gravity wave disturbance, near 10 km in the troposphere and near the tropopause at 20 km, with values around -3 K and 1 K. The maximum at 10 km reflects the role of the jet stream, and that at 20 km reflects the role of the tropopause. It can be seen from Fig.



4a that above the tropopause from 18 to 20 km, the background temperature reaches a minimum in the
vertical direction, and the vertical temperature lapse rate is also very large. At the same time, Figure 1b
shows that the gravity wave disturbance also reaches a maximum near this height range. According to
energy conservation, the sharp increase in the amplitude of the gravity wave disturbance in this height
range cannot be physical. Above 20 km, there is a significant wave structure in the vertical direction.
The wavelengths of the gravity waves above 20 km are more than 8 km, according to the vertical
sliding average method. Below the tropopause, waves with smaller scales of 5–7 km are included.
It should be noted that the sudden temperature changes with height near the tropopause cannot be
smoothed out using the vertical sliding average method. It is speculated that the calculated gravity
wave disturbance is amplified artificially by this method, resulting in an error. However, this error only
occurs near the tropopause: above 20 km, because the vertical variation of temperature is relatively flat,
the neighboring temperatures contributing to the atmospheric background temperature are reasonably
representative of that height, and the calculated gravity wave disturbance is therefore also highly
reliable.

As is well known, the square of the buoyancy frequency represents the characteristics of the
background atmosphere, while the gravity wave potential energy profile represents the transient
behavior. From the calculated gravity wave disturbance, the square of the background atmospheric
buoyancy frequency and the gravity wave potential energy profile are calculated by using Eqs. (1) and
(2), and shown in Figs. 1c and 1d, respectively. The tropospheric buoyancy frequency gradually
increases with height. Above the tropopause, the buoyancy frequency reaches a maximum value, which
is similar to the results given by the previous study (Liao et al., 2016). The variation of the buoyancy
frequency is related to the potential energy of the gravity wave. The potential energy of the gravity
wave in the troposphere gradually decreases and reaches a minimum near the tropopause for the first
time.

Although the gravity wave disturbance near the tropopause is amplified when the atmospheric
background temperature profile is extracted using the vertical sliding average method, some
components of the small-vertical-scale disturbances are retained: As well as fluctuations in the vertical
with wavelengths 5–7 km, smaller-scale disturbances can also be seen.

Generally, the vertical sliding average method appears to retain high temporal resolution, because of
the preservation of the atmospheric temperature profile at each time in the calculation. However,
accurate gravity wave disturbance profiles are not available, mainly for two reasons: (1) near the
tropopause, the gravity wave disturbance is artificially overestimated; (2) small-scale gravity waves
and small-scale turbulence cannot be distinguished.




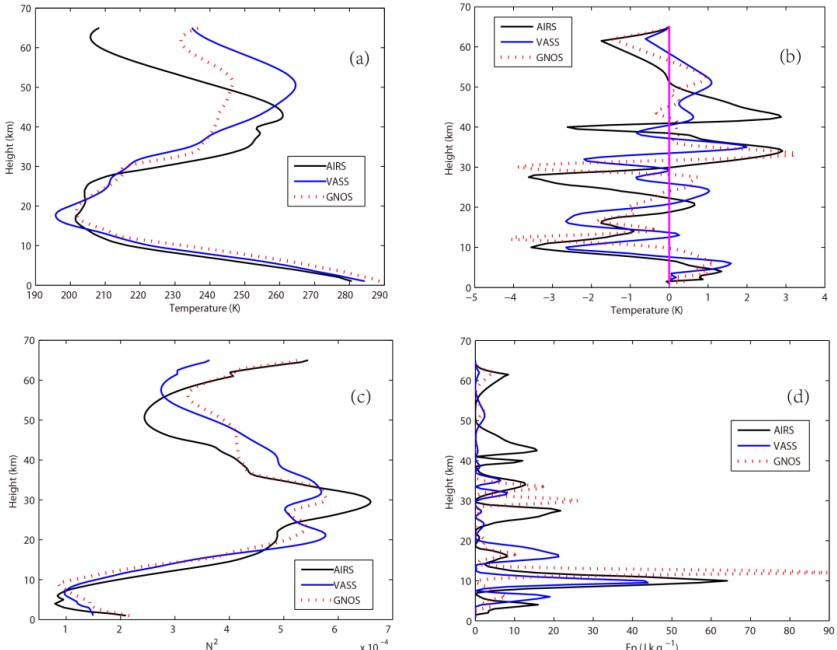

**Figure 2.** (a) Temperature profile at 74.65ºW, 35.19ºN on January 1, 2019. (b) The gravity wave temperature disturbance profile. (c) The gravity wave buoyancy frequency. (d) The gravity wave potential energy. Quantities are calculated using the vertical sliding average method. Black lines indicate AIRS, blue lines VASS, and red dotted lines GNOS.

The calculation results from applying the vertical sliding average method to of FY-3 satellite and AIRS data are compared in Fig. 2. On the whole, the three data sets can reflect the temperature variations. Below 20 km, the temperature profile of the three data sets is consistent (Fig. 2a). From 20 km to 35 km, the difference is gradually becomes larger. The VASS temperature profile includes global atmospheric temperature profiles retrieved from 4 MWTS (Micro Wave Temperature Sounder) microwave channels. Less energy is received near the ground, so the VASS temperature is of lower quality here. Above 35 km, the temperature profiles of GNOS and AIRS are very different. This is because the GNOS SNR (Signal to Noise Ratio) decreases above 35 km: it generates some false signals, which reduce the quality of the temperature profile, showing that GNOS is applicable in the range 5–35 km. From 35 km to 50 km, the temperature profile of VASS is more consistent with that of AIRS than that of GNOS. Additionally the results for VASS are improved because its infrared spectrometer can receive more radiation signals in this height range. But because it's lower vertical resolution above 30 km, the height of the maximum is not accurate.

In the height range 5–35 km, the gravity wave disturbances from GNOS are consistent with AIRS,



although the amplitude from GNOS is larger, up to 4 K near the tropopause (Fig. 2b). Within this height range, GNOS has an even stronger gravity wave signal than AIRS. And some components of the

small-vertical-scale disturbances are retained: As well as fluctuations in the vertical with wavelengths 5–7 km, smaller-scale disturbances can also be seen.

The square of the buoyancy frequency and the gravity wave potential energy are compared in Figs. 2c and 2d, respectively. Although both GNOS and VASS follow similar behavior to AIRS, they have their own advantages. Below 35 km, GNOS shows a stronger signal than VASS and AIRS, for both

buoyancy frequency and gravity wave potential energy. With increasing height, the false signal from GNOS increases while VASS can obtain more radiant energy, so that the accuracy of VASS gradually increases relative to that of GNOS. But because of the lower vertical resolution of VASS above 30 km, the height of the maximum is not accurate.

From the above results it can be concluded that, in the height range 5–35 km, the gravity wave signal

obtained by GNOS is better, and some components of the small-vertical-scale disturbances are retained: As well as fluctuations in the vertical with wavelengths 5–7 km, smaller-scale disturbances can also be seen. The vertical resolution of VASS is lower, but larger vertical scale components are retained. The sliding average method can be applied at each height of a single temperature profile to obtain a rough background temperature. But the temperature disturbance based on this background temperature

calculation method will inevitably contain some small-scale disturbances in the vertical direction, as well as other fluctuations, that are comparable in vertical scale to gravity waves, but differ in temporal scale, for example planetary waves. In addition, this method cannot accommodate drastic changes in background temperature in the vertical direction. For example, near the tropopause, the gravity wave disturbance potential energy is overestimated.

**3.2 Double-filter method**

In order to avoid the overestimation of the gravity wave disturbance near the tropopause from using the vertical sliding average method, the double-filter method is adopted. The steps are as follows:

(1) In the region 30°N–40°N, 70°W–90°W, from January 1 to 7, 2019, the average temperature at each height is calculated and used as the background $\bar{T}$.

(2) The observed temperature profile at (74.65°W, 35.19°N) on January 1, 2019, is taken as the original profile; the background is subtracted to obtain the disturbance profile.

(3) Linear fitting of the disturbance profile in the vertical direction is applied and the vertical linear trend is removed.

(4) A high-pass filter with a vertical wavelength of 10 km is applied.

(5) A low-pass filter with a vertical wavelength of 2 km is applied, to finally obtain the gravity wave disturbance profile.

When using this method, the gravity wave disturbance obtained contains wave components with vertical wavelengths from 2 km to 10 km. The calculation results for AIRS are given in Fig. 3, and then the calculation results from the FY-3 satellite and AIRS are compared in Fig. 4.


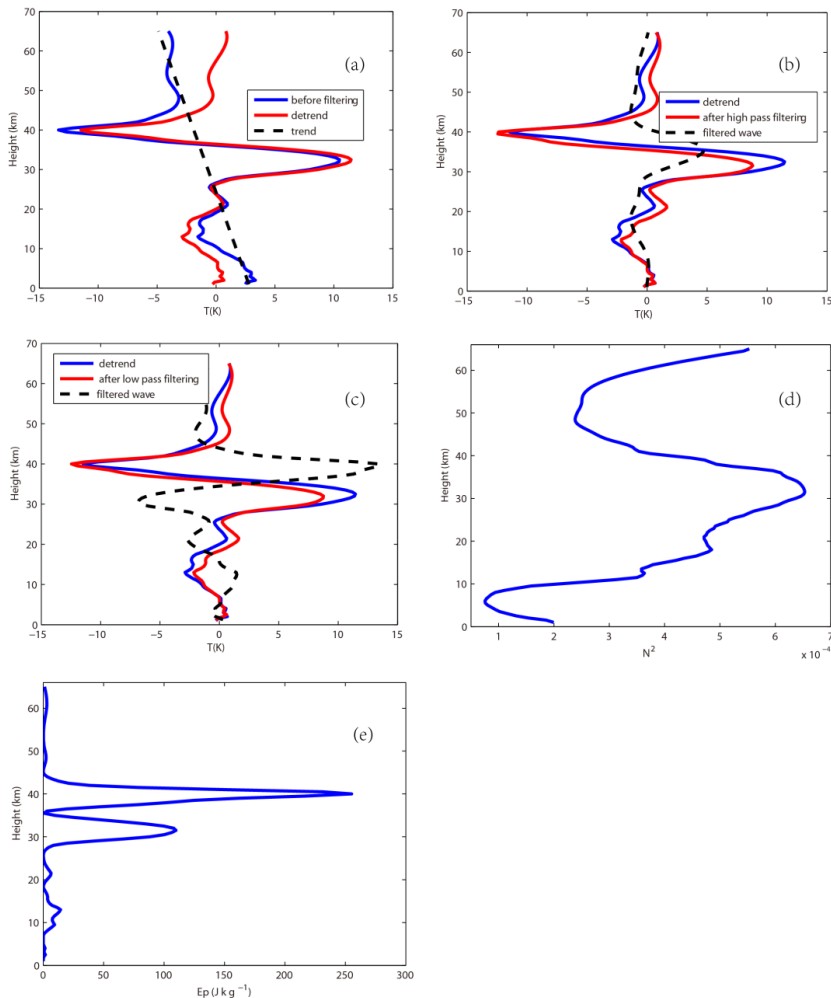

**Figure 3.** Results from the double-filter method. The physical quantities expressed in each panel refer to the descriptions in the text.

Figure 3a shows the disturbance profile (blue line) obtained by subtracting the background from the original profile. It can be seen that the disturbance profile has a significant linear trend in the vertical direction, and the trend is clearly not caused by any gravity wave disturbance, so the vertical linear trend (black dotted line) is removed. The red line in Figs. 3a, 3b and 3c is the temperature disturbance





profile after removing the linear trend. From this, the filtering method is used to extract the gravity

wave disturbance from a single disturbance profile. First, a high-pass filter with a vertical wavelength
of 10 km is used to filter out large vertical disturbances, such as planetary waves. The results of the
high-pass filter are shown in Fig. 3b. The red line is the same as in Fig. 3a, the blue line represents the
profile after the high-pass filtering, and the black dashed line represents the large-scale disturbances,
with vertical wavelength greater than 10 km, filtered out by the high-pass filtering. The calculation

result of the low-pass filtering is shown in Fig. 3c. The red line is the same as in Fig. 3a, the black
dotted line indicates the small-scale disturbances, with vertical wavelength less than 2 km, filtered out
by the high-pass filtering, and the blue line indicates the final gravity wave disturbance profile obtained
after the low-pass filtering.

Comparing the gravity wave disturbance profile (blue line in Fig. 3c) with the results in Fig. 1b (blue

line), it can be seen that the gravity wave disturbance obtained by the vertical sliding average method
has a richer vertical variation. This shows that the vertical sliding average method retains some small
vertical scale disturbances: as well as fluctuations with vertical wavelengths of 5–7 km, smaller-scale
disturbances can be seen. The gravity wave disturbance profile obtained by the double-filter method
contains fewer small disturbances, and the main gravity wave has a vertical wavelength of about 8 km

(blue line in Fig. 3c), which is similar to the results given by the previous study (Wang et al., 2019).
Based on the calculated gravity wave disturbance, the square of the buoyancy frequency and the gravity
wave potential energy profile are shown in Figs. 3d and 3e, respectively. The buoyancy frequency
gradually increases with height in the troposphere. Above the tropopause, the buoyancy frequency
reaches a maximum value, which is similar to the results given by the previous study (Liao et al., 2016;

Wang et al., 2019). The variation of the buoyancy frequency is also related to that of the potential
energy of the gravity wave.

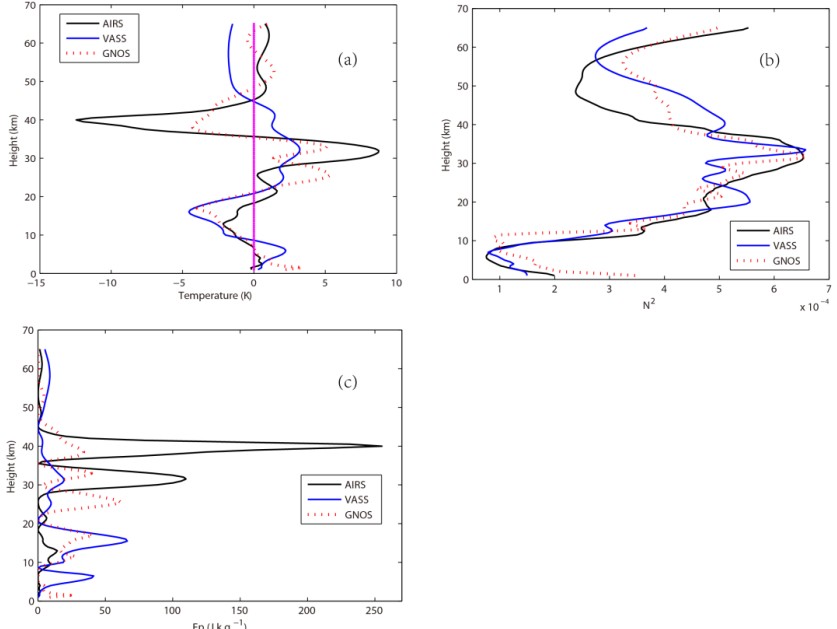

**Figure 4.** (a) Gravity wave temperature disturbance profile. (b) Gravity wave buoyancy frequency. (c)
Gravity wave potential energy. Quantities are calculated using the double-filter method. Black lines
indicate AIRS, blue lines VASS, and red dotted lines GNOS.

The results of the double-filter method for extracting gravity waves are obtained based on AIRS. Then
the calculation results from applying the double-filter method to FY-3 satellite and AIRS profiles are
compared in Fig. 4.

Comparing the gravity wave disturbance profiles (Fig. 4a), in the height range 5–35 km, the gravity
wave disturbance from GNOS is more consistent with that from AIRS. Meanwhile, AIRS and GNOS
reflect some small-scale fluctuation information, with a wavelength of about 3–5 km, which VASS is
unable to obtain. Above 35 km, AIRS can obtain more radiant energy, so that AIRS has an even
stronger gravity wave signal than GNOS. However, because of the lower vertical resolution of VASS
above 30 km, the accuracy of VASS gradually decreases relative to that of GNOS and AIRS. Those are
also seen with the calculated buoyancy frequency (Fig. 4b) and potential energy (Fig. 4c). As with the
vertical sliding average method, from 5 km to 35 km, GNOS has an even stronger gravity wave signal
than AIRS. Above 35 km, the gravity wave disturbance of AIRS is stronger.

In conclusion, in the height range 5–35 km, the gravity wave signal obtained by GNOS is better, and in
the range 35–65 km, the gravity wave signal obtained by AIRS is better. The gravity wave disturbance
extracted by the double-filter method is generally very accurate. The double-filter method can





effectively suppress the large-scale background and small-scale disturbances in the temperature profile, so the obtained profile represents the sum of gravity wave disturbances with vertical wavelengths of 2–

10 km. According to these results, the gravity wave potential energy can be calculated more accurately. However, although the gravity wave potential energy at a certain height can be calculated accurately, the variation of the gravity wave potential energy with height is not so well reflected. But since this is not essential to the arguments developed in this article, it will not be pursued further here.

### 3.3 Single-filter method

Taking the observed temperature profile at 74.65°W, 35.19°N on January 1, 2019, as an example, the single-filter method is used to extract the gravity wave disturbance profile from a single temperature profile. The steps are as follows:

(1) Temperature profiles are obtained in the region 30°N–40°N, 70°W–90°W, from January 1 to 7, 2019.

(2) Each profile is interpolated onto a vertical grid with a spacing of 1 km. This is equivalent to low-pass filtering in the vertical direction, filtering out disturbances and noise in the temperature profile with vertical wavelengths less than 1 km.

(3) Within the region 30°N–40°N, 70°W–90°W, from January 1 to 7, 2019, the average temperature in each height is calculated and used as the background $\bar{T}$.

(4) The observed temperature profile at 74.65°W, 35.19°N on January 1, 2019, is taken as the original profile, and the background profile is subtracted to obtain the disturbance profile.

(5) A high-pass filter with a vertical wavelength of 8 km is used to filter the disturbance profile. Finally, the gravity wave disturbance profile is obtained containing wavelengths less than 8 km.



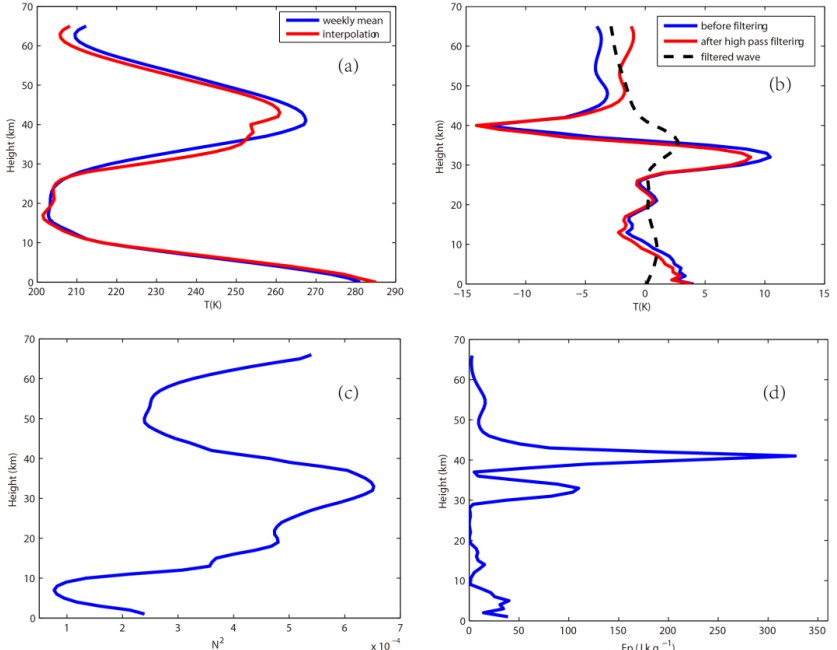


**Figure 5.** Results from the single-filter method. The physical quantities expressed in each panel refer to the descriptions in the text.

The calculation results for AIRS are given in Fig. 5, and the calculation results for the FY-3 satellite

and AIRS are compared in Fig. 6.

In Fig. 5a, the red line represents the temperature profile after interpolation, and the blue line represents the 7-day average temperature profile. This shows that the interpolation in the vertical direction weakens the sharp vertical variation of the atmospheric temperature near the tropopause, which reduces the error caused by the calculation for extracting the gravity wave disturbance. However, the calculated

disturbance profile still has a significant linear trend in the vertical direction, since only the time average is removed. A high-pass filter can be used to remove wave components with large vertical scales.

The result of the high-pass filtering is shown in Fig. 5b. The blue line in the figure is the temperature disturbance profile before filtering, the red line is the gravity wave temperature disturbance profile after

filtering, and the black dashed line is the large-scale background removed by the filter. The vertical variation trend of the obtained gravity wave disturbance is consistent with the results obtained using the double-filter method in Fig. 3c. Overall, there are little differences in the absolute value and vertical wavelength of the gravity wave disturbance obtained by the two methods. The absolute value of gravity wave disturbance obtained by the single-filter method is similar to that obtained by the double-filter

method. Moreover, there are also small-scale vertical disturbances for the single-filter method. It also

shows that there are many small-vertical-scale gravity waves in the upper troposphere and lower

stratosphere.

Based on the calculated gravity wave disturbance, the square of the buoyancy frequency and the gravity

wave potential energy profiles are shown in Figs. 5c and 5d, respectively. The vertical trend and

magnitude of $N^2$ are generally consistent with the results of Fig. 3d. The variation of the buoyancy

frequency is also related to that of the potential energy of the gravity wave e, which is similar to the

results given by the previous study (Liao et al., 2016; Wang et al., 2019).

The results of the single-filter method for extracting gravity waves, for the FY-3 satellite and AIRS, are

compared in Fig. 6.


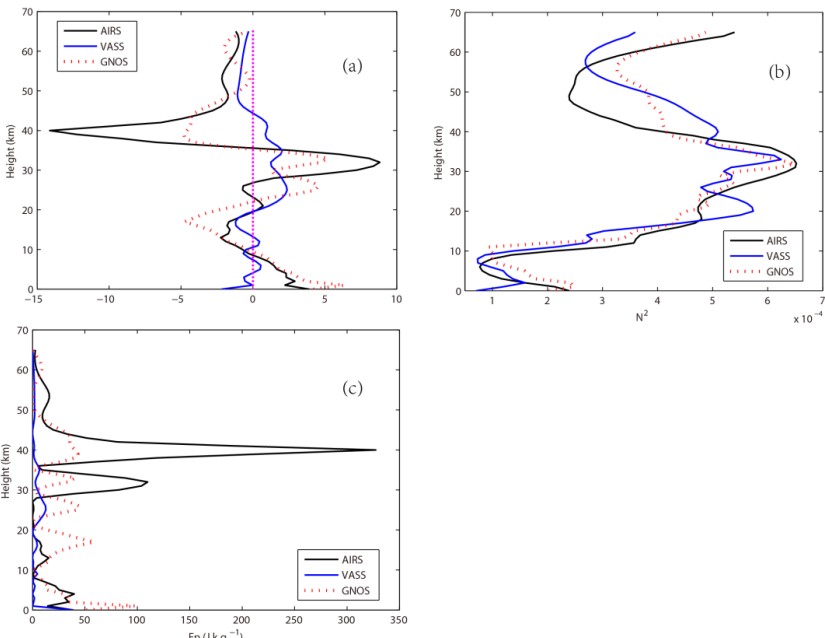

**Figure 6.** (a) Gravity wave temperature disturbance profile. (b) Gravity wave buoyancy frequency. (c)
Gravity wave potential energy. Quantities are calculated using the single-filter method. Black lines
indicate AIRS, blue lines VASS, and red dotted lines GNOS.


Comparing the gravity wave disturbance profiles (Fig. 6a), in the height range 5–35km, the gravity

wave disturbance profile from GNOS is more consistent with that from AIRS. Meanwhile, AIRS and

GNOS also reflect some small-scale fluctuation information, with wavelengths of about 3–5 km. After

the high-pass filtering, the vertical linear trend in the disturbance is filtered out well. The vertical



variation of the obtained gravity wave disturbance is consistent with the results obtained using the
double-filter method. Above 35 km, the amplitude of the gravity wave disturbance from AIRS is larger
than those from GNOS and VASS, reaching more than 8–10 K and 10-10K, with large height variation
of VASS. As with the first two methods, from 5 km to 35 km, GNOS has an even stronger gravity wave
signal than AIRS. Above 35 km, after the high-pass filtering, the vertical linear trend in the disturbance
from AIRS is filtered out well. And this is also the case for the variation of the buoyancy frequency
(Fig. 6b) and the potential energy of the gravity wave (Fig. 6c).

In general, in the height range 5–35 km, the gravity wave signal obtained by GNOS is better, and in the
range 35–65 km the gravity wave signal obtained by AIRS is better. Because of the lower vertical
resolution of VASS, the gravity wave signal obtained by VASS is different from AIRS and GNOS. The
gravity wave disturbance extracted by the single-filter method is generally very accurate. The gravity
wave disturbance obtained using the single-filter method essentially does not contain a large-scale
background in the vertical direction, but still contains some small-vertical-scale disturbances.

**4 Conclusions**

In order to further investigate the advantages and disadvantages of FY-3, the two types of temperature
profile products from FY-3, with GNOS and VASS, together with AIRS operational Level 2 data, are
used to compare and analyze the gravity wave parameter based on three extraction methods. The main
results are as follows:

1. The results calculated by the three methods are generally consistent. However, they have advantages
and disadvantages. First, the vertical sliding average method can extract gravity wave disturbances with
a vertical wavelength of less than8 km. However, because of the overestimation of gravity wave
disturbance near the tropopause, accurate gravity wave disturbance profiles are not available.

2. Second, the double-filter method can extract gravity waves with a vertical wavelength range from 2
to 10 km. The gravity wave disturbance value obtained by this method is more accurate.

3. Third, the single-filter method can obtain gravity wave disturbances in the vertical wavelength range
from 2 to 8 km, but the gravity wave disturbance profiles obtained using this method still contain some
small-vertical-scale disturbances. Due to linear interpolation, each profile resolution reduces to 1 km.

4. In comparing the three gravity wave parameter extraction methods, it is found that the GNOS
temperature profile product has better results in the lower layer of 5–35 km. From 35 to 65 km AIRS is
better than GNOS. From 5 to 35 km, when the double-filter or single-filter method is used, GNOS
contains small-scale information. Using the single-filter method, GNSS and AIRS filter out the vertical
linear trend in the disturbance profile well, reflecting an advantage of this method. The vertical
resolution of VASS is lower, but larger vertical scale components are retained.





*Data Availability*. The data used in this paper are available from the corresponding author upon request.


*Author contributions*. The central idea is mainly contributed by ZS and SC. The methods used in this manuscript are conceived by ZS and WG. The retrieval algorithm is developed SC, ZS and WG. The results are discussed by SC, ZS and WG. SC analyzed the data, prepared the figures and wrote the paper. WZ, YH and ZL contributed to refining the ideas, carrying out additional analyses. All

co-authors reviewed the paper.

*Competing interests*. The authors declare that they have no conflict of interest.

*Acknowledgements*. The study was supported by the National Key R&D Program of China (No.

2017YFC1501802 and No. 2018YFA0605604) and the National Natural Science Foundation of China (Grant no. 41875045).

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
