# Peer review of "Intercomparison of FY-3 and AIRS Gravity Wave Parameter Extraction Based on Three Methods"

_Annales Geophysicae, 2019_

## Referee Comment (RC1) · Anonymous Referee #1 · 6 Oct 2019

General comments:

This work investigates the advantage and disadvantage of three types of temperature profile data for gravity wave parameter extraction based on three extraction methods, and some results which have potential in application are obtained. The logical structure of this manuscript is good. This work does not involve complex scientific issues (such as physical mechanisms), but mainly focuses on the application of technology. Therefore, the authors should concentrate on describing the results accurately. There are some grammatical and typographical errors in the manuscript. Besides, many sentences are not very clear and professional. These should be corrected before the manuscript to be published. I only list small part of errors as below, hope the authors will examine the article carefully.

[Figure]

1. AIRS as a nadir sounder can only observe gravity waves with long vertical wavelengths. Hoffmann and Alexander (2009) found that the minimum vertical wavelength of their high-resolution AIRS retrieval is about 10-15 km in the stratosphere. The authors need to further indicate that AIRS Level 2 product can be used to extract gravity wave.

2. Many proper nouns are abbreviated when they first appear. Please check.

3. L53: prediction mode-> prediction model

4. L98: which showed-> which showed. An extra blank.

5. L122: "...between 1100 and .016 mb" What's the meaning of ".016 mb"? Please make it clear.

6. L211-L213: "In fact, some planetary scale disturbances, such as Kelvin waves, have vertical wavelengths of the same scale as gravity waves, so using these filtering methods means that the gravity wave disturbances obtained actually include a contribution from these other waves." "in fact" and "actually" are completely repetitive.

7. L296-L297: From 20 km to 35 km, the difference is gradually becomes larger. -> From 20 to 35 km, the difference gradually becomes larger.

8. L362: with vertical wavelength less than 2 km ->with a vertical wavelength less than 2 km.

9. Note that the figures of GW properties would be compared with the previous literatures.

---

## Referee Comment (RC2) · Anonymous Referee #2 · 10 Oct 2019

This paper introduced gravity wave extractions from temperature profiles of three measurements, AIRS, GNOS, VASS. Three methods were employed: vertical sliding average, double-filter and single-filter.

Overall, the English in the paper is not accurate and scientific at all, making the review task very difficult. The context does not warrant a publication in AG with little physics or any new insights on gravity waves. The filtering and background removing are very common techniques to analyze GWs in temperature profiles. Unfortunately, I would recommend rejection of this paper.

Some detailed comments:

1. line 24: "the GNOS temperature profile product performs better..." What does "bet-

ter" mean here? Should be more specific. This problem is all over the paper. Line 25: " the AIRS temperature profile product is better than GNOS".

2. How the weighting function of AIRS impacts the vertical resolution of temperature profiles?

3. Is GNOS radio occultation? If so, would COSMIC a better validation?
* * *

---

## Author Comment (AC1) · 10 Oct 2019

Referee #1 We thank Referee #1 for his/her comments and useful remarks. In the following we include our answers point-by-point.

1. AIRS as a nadir sounder can only observe gravity waves with long vertical wavelengths. Hoffmann and Alexander (2009) found that the minimum vertical wavelength of their high-resolution AIRS retrieval is about 10-15 km in the stratosphere. The authors need to further indicate that AIRS Level 2 product can be used to extract gravity wave. Yes, you are right. In this manuscript we have discussed this point according to the article (Hoffmann and Alexander, 2009) (L106-L123). The AIRS operational Level 2 data is used in this paper which has been analyzed in Section 2 (L139-L149). In order

to further indicate that AIRS Level 2 product can be used to extract gravity wave, the missing reference has been added to this part. It can be seen as follows: "AIRS operational Level-2 data provides stratospheric temperature profiles. For validation AIRS operational Level-2 data are compared with results from the ERA-Interim meteorological reanalysis. The comparisons show that AIRS operational Level-2 temperature data are in good agreement with the validation data sets, and are considered optimal for gravity wave studies (Meyer and Hoffmann, 2014)." (L144-L149) "Meyer, C. I., and Hoffmann, L.: Validation of AIRS high-resolution stratospheric temperature retrievals, Proc. SPIE 9242, Remote Sensing of Clouds and the Atmosphere XIX; and Optics in Atmospheric Propagation and Adaptive Systems XVII, 92420L, https://doi.org/10.1117/12.2066967, 2014." (L606-L608) Thanks.

2. Many proper nouns are abbreviated when they first appear. Please check. Sorry for my carelessness. In abstract: "Abstract. Two types of temperature profile products from the FY-3 (FengYun-3) satellite system, using GNOS and VASS, together with AIRS operational Level 2 data, are used to compare and analyze gravity wave parameters." This has been modified to "Abstract. Two types of temperature profile products from the FY-3 (FengYun-3) satellite system, using GNOS (The Global Navigation Occultation Sounder) and VASS (The Vertical Atmospheric Sounder System), together with AIRS (The Atmospheric Infrared Sounder) operational Level 2 data, are used to compare and analyze gravity wave parameters." (L 15-L 18)

"Subsequently, the global distribution characteristics of stratospheric gravity waves in a given year have been studied by using GPS/MET occultation data (Tsuda et al., 2000), CHAMP GPS occultation data (Ratnam et al., 2004; Torre et al., 2006), Aura satellite MLS (Microwave Limb Sounder) observations (Wu and Eckermann, 2008) and COSMIC GPS occultation data (Xiao and Hu, 2010)." This has been modified to "Subsequently, the global distribution characteristics of stratospheric gravity waves in a given year have been studied by using GPS/MET (The Global Positioning System/Meteorology) occultation data (Tsuda et al., 2000), CHAMP (The Challenging Minisatellite Payload) GPS occultation data (Ratnam et al., 2004; Torre et al., 2006), Aura satellite MLS (Microwave Limb Sounder) observations (Wu and Eckermann, 2008) and COSMIC (Constellation Observation System for Meteorology, Ionosphere and Climate) GPS occultation data (Xiao and Hu, 2010)." (L 71-L 76)

"In order to construct a more systematic and reliable gravity wave model, Ern et al. (2014) used SABER's 11-year observation data and HIRDLS (High Resolution Dynamics Limb Sounder) two-year observation data to study the contribution of gravity waves to the equatorial quasi-biennial oscillation (QBO)." This has been modified to "In order to construct a more systematic and reliable gravity wave model, Ern et al. (2014) used SABER (Sounding of the Atmosphere Using Broadband Emission Radiometry) 11-year observation data and HIRDLS (High Resolution Dynamics Limb Sounder) two-year observation data to study the contribution of gravity waves to the equatorial quasi-biennial oscillation (QBO)." (L 78-L 81)

"The two types of FY-3 temperature profile products, with GNOS and VASS, together with AIRS operational Level 2 data, are used to compare and analyze gravity wave parameters." This has been modified to "The two types of FY-3 temperature profile products, with GNOS (The Global Navigation Occultation Sounder) and VASS (The Vertical Atmospheric Sounder System), together with AIRS operational Level 2 data, are used to compare and analyze gravity wave parameters." (L 136-L 139)

"The Level 2 data of FY-3 include atmospheric temperature profiles from the Global Navigation Occultation Sounder (GNOS) and the Vertical Atmospheric Sounder System (VASS) (Liao et al., 2016; Yao et al., 2019)." This has been modified to "The Level 2 data of FY-3 include atmospheric temperature profiles from GNOS and VASS (Liao et al., 2016; Yao et al., 2019)." (L 161-L 162) Thanks.

3. L53: prediction mode-> prediction model Yes, you are right. This has been modified. (L 54) Thanks.

4. L98: which showed-> which showed. An extra blank. Yes, you are right. This has

been modified. (L 102) Thanks.

5. L122: "...between 1100 and .016 mb" What's the meaning of ".016 mb"? Please make it clear. Sorry about this. This has been modified to "...between 1100 and 0.016 mb" (L126) Thanks.

6. L211-L213: "In fact, some planetary scale disturbances, such as Kelvin waves, have vertical wavelengths of the same scale as gravity waves, so using these filtering methods means that the gravity wave disturbances obtained actually include a contribution from these other waves." "in fact" and "actually" are completely repetitive. Yes, we agree with you. This has been modified to "In fact, some planetary scale disturbances, such as Kelvin waves, have vertical wavelengths of the same scale as gravity waves, so using these filtering methods means that the gravity wave disturbances obtained include a contribution from these other waves." (L 218-L 220) Thanks.

7. L296-L297: From 20 km to 35 km, the difference is gradually becomes larger. -> From 20 to 35 km, the difference gradually becomes larger. Yes, we agree with you. This has been modified. (L 304) Thanks.

8. L362: with vertical wavelength less than 2 km ->with a vertical wavelength less than 2 km. Sorry about this. The corresponding sentences have been modified as follows: "...with vertical wavelengths greater than 10 km" (L 366) "...with vertical wavelengths less than 2 km" (L 368) Thanks.

9. Note that the figures of GW properties would be compared with the previous literatures. Yes, you are right. The comparisons to the previous literature have been made. It can be seen as follows: Wang et al., 2019 (L 268). Liao et al., 2016 (L 282). Liao et al., 2016; Wang et al., 2019 (L382-383; L455).

Thanks again for your careful review. Hopefully our response can enable a further review of the manuscript. We will fix all these points in the final version following your suggestions. Many thanks for your work so far and best regards, Shujie Chang and
Co-authors. Please also note the supplement to this comment.

Please also note the supplement to this comment:
https://www.ann-geophys-discuss.net/angeo-2019-130/angeo-2019-130-AC1-supplement.zip

---

## Author Comment (AC2) · 18 Oct 2019

Referee #2 We thank Referee #2 for his/her comments and useful remarks. In the following we include our answers point-by-point.

General comments Overall, the English in the paper is not accurate and scientific at all, making the review task very difficult. Yes, you are right. I apologize for the confusion that the English in the paper is not accurate and scientific. This, as you pointed out, may cause the readers hard to understand. Following your suggestion, in order to increase the readability of the paper, foreign experts have been asked to do further modify this manuscript. Thanks. The context does not warrant a publication in AG with little physics or any new insights on gravity waves. The filtering and background removing are very common techniques to analyze GWs in temperature profiles. Sorry for this. This work does not involve complex scientific issues (such as physical mechanisms), but mainly focuses on the application of technology. This study is aimed to examine the applicability of the latest Chinese satellite observations in gravity wave studies. The most difficult aspect of this work is the use of Chinese satellite observations. The precious first-hand data from Chinese advanced satellites has been rarely used for further studies. This study introduces the Chinese second generation satellite observed data and compares it with those well-known data set, which will help those studies based on observational data. We have modified the concentration on this. The abstract has been modified to "Abstract. Two types of temperature profile products from the FY-3 (FengYun-3) satellite system, GNOS (The Global Navigation Occultation Sounder) and VASS (The Vertical Atmospheric Sounder System), together with AIRS (The Atmospheric Infrared Sounder) operational Level 2 temperature profile product, are used to compare and analyze gravity wave parameters. The advantages and disadvantages of these three types of temperature profile data for gravity wave parameter extraction are determined, based on three extraction methods: vertical sliding average, double-filter and single-filter. By comparing the three methods, the conditions under which each dataset can be applied are obtained. Accurate gravity wave disturbance profiles cannot be obtained using the vertical sliding average method. The double-filter method can extract gravity waves with wavelengths of 2 to 10 km. The single-filter method can obtain gravity wave disturbances with vertical wavelengths less than 8 km. For all three gravity wave parameter extraction methods, the GNOS temperature profile product shows a stronger signal than VASS and AIRS, for both buoyancy frequency and gravity wave potential energy in the lower layer (5–35 km). From 35 to 65 km the gravity wave signal obtained by AIRS is better than GNOS. The vertical resolution of VASS is lower, but larger vertical scale components are retained." (L 15- L 29) In section 1 introcustion "Atmospheric gravity waves are small-scale or meso-scale disturbances that can propagate vertically (Holton, 1983)......Second, considering the subgrid effects of stratospheric gravity waves is important for constructing the parameterization

scheme itself (Fritts and Alexander, 2003; Kim et al., 2010)." has been modified to "Atmospheric gravity waves are small-scale or meso-scale disturbances that can propagate vertically (Holton, 1983)......On the one hand, it is essential for improving the accuracy of atmospheric circulation models and the numerical weather prediction; on the other hand, it is an urgent requirement for flight safety."(L 32- L 62) "During aircraft flight, since the scale of the gravity waves is similar to the typical aircraft size, stratospheric gravity waves have a strong influence on the aircraft, and can periodically cause it to vibrate......" has been modified to "By using data obtained from various observation methods, information about stratospheric gravity waves can be extracted, and their distribution characteristics analyzed....... In summary, it is possible to extract good gravity wave signals from temperature profiles retrieved by AIRS." (L 63- L101) "This study is aimed to examine...which will help those studies based on observational data." a new paragraph has been added. (L 132 –L 138) In section 2 "2.1 AIRS Level 2 data" has been modified to "2.1 AIRS Level 2 temperature profile product" (L 140) "2.2 FY-3 temperature profile" has been modified to "2.2 FY-3 temperature profile product" (L 165) In conclusions, "In order to further investigate the advantages and disadvantages of FY-3......The vertical resolution of VASS is lower, but larger vertical scale components are retained." has been modified to "This study examines the applicability of the latest Chinese satellite observations in gravity wave......but larger vertical scale components are retained." (L 484- L 504) Thanks Some detailed comments: 1. line 24: "the GNOS temperature profile product performs better..." What does "better" mean here? Should be more specific. This problem is all over the paper. Line 25: "the AIRS temperature profile product is better than GNOS". Yes, you are right. Those have been modified. "...the GNOS temperature profile product performs better in the lower layer of 5–35 km" has been modified to "the GNOS temperature profile product shows a stronger signal than VASS and AIRS, for both buoyancy frequency and gravity wave potential energy in the lower layer (5–35 km)." (L 25-L 27) "the AIRS temperature profile product is better than GNOS" has been modified to "From 35 to 65 km the gravity wave signal obtained by AIRS is better than GNOS" (L 27- L 28) "the temperature profiles

here are poor" has been modified to "the temperature profiles here are inaccurate" (L 261) "…the height of the maximum is not accurate" has been modified to "…the height of the maximum fluctuation is not accurate" (L 316) "…the height of the maximum is not accurate." has been modified to "…the height of the maximum fluctuation is not accurate."(L 328) "AIRS is better than GNOS" has been modified to "the gravity wave signal obtained by AIRS is better than GNOS" (L500) Thanks 2. How the weighting function of AIRS impacts the vertical resolution of temperature profiles? Sorry for the confusion. Now, this has been introduced in section 2 "The weighting functions are used to transform correlative measurements to AIRS effective resolution and are used to assess and derive the vertical resolution of temperature and moisture retrievals in different atmospheric conditions (Maddy and Barnet, 2008)……(More details can be seen at https://disc.gsfc.nasa.gov/ datasets/AIRS2RET_NRT_006/summary, 2019)." (L 151-L 158) "Maddy, E.S., and Barnet, C.D.: Vertical resolution estimates in version 5 of AIRS operational retrieval, IEEE TGARS, 46, 2375–2384, doi: 10.1109/TGRS.2008.917498, 2008." (L 607-L 608) Thanks. 3. Is GNOS radio occultation? If so, would COSMIC a better validation? Sorry for the confusion. GNOS is radio occultation which is one of the remote sensing instruments on the FY-3 satellite. If the study is aimed to examine just the applicability of GNOS in gravity wave studies, COSMIC would a better validation. However, the study is not only aimed to examine the applicability of GNOS, but VASS in gravity wave studies, both of which are the latest Chinese satellite observations. The AIRS covers a wide band of observation the brightness temperature: 3.74 $\mu$m to 4.61 $\mu$m, 6.20 $\mu$m to 8.22 $\mu$m, and 8.8 $\mu$m to 15.4 $\mu$m, totally in 2378 channel. Comparing with other hyperspectral measurements, AIRS contains more channels, which forms a high spatial resolution.

Thanks again for your careful review. Hopefully our response can enable a further review of the manuscript. We will fix all these points in the final version following your suggestions. Many thanks for your work so far and best regards, Shujie Chang and Co-authors. Please also note the supplement to this comment.

[Figure]

Please also note the supplement to this comment:
https://www.ann-geophys-discuss.net/angeo-2019-130/angeo-2019-130-AC2-supplement.zip
* * *